# Attachment performance of the ectoparasitic seal louse *Echinophthirius horridus*

Anika Preuss [1✉], Thies H. Büscher [1], Insa Herzog [2], Peter Wohlsein[3], Kristina Lehnert[2] & Stanislav N. Gorb [1]

Marine mammals host a great variety of parasites, which usually co-evolved in evolutionary arms races. However, little is known about the biology of marine mammal insect parasites, and even less about physical aspects of their life in such a challenging environment. One of 13 insect species that manage to endure long diving periods in the open sea is the seal louse, *Echinophthirius horridus*, parasitising true seals. Its survival depends on its specialised adaptations for enduring extreme conditions such as hypoxia, temperature changes, hydrostatic pressure, and strong drag forces during host dives. To maintain a grip on the seal fur, the louse's leg morphology is equipped with modified snap hook claws and soft pad-like structures that enhance friction. Through techniques including CLSM, SEM, and histological staining, we have examined the attachment system's detailed structure. Remarkably, the seal louse achieves exceptional attachment forces on seal fur, with safety factors (force per body weight) reaching 4500 in average measurements and up to 18000 in peak values, indicating superior attachment performance compared to other insect attachment systems. These findings underscore the louse's remarkable adaptations for life in a challenging marine environment, shedding light on the relationship between structure and function in extreme ecological niches.

[1] Department of Functional Morphology and Biomechanics, Zoological Institute, Kiel University, Kiel, Germany. [2] Institute for Terrestrial and Aquatic Wildlife Research, University of Veterinary Medicine Hannover, Büsum, Germany. [3] Department of Pathology, University of Veterinary Medicine Hannover, Hannover, Germany. ✉email: apreuss@zoologie.uni-kiel.de

Insects are the most successful animal class regarding both the number of species and habitat diversity[1]. Although they are the most common and widespread terrestrial animals, only very few species live in the sea, due to the challenges this habitat poses to its flora and fauna[2]. It is therefore not surprising that there is only one single insect lineage with 13 species that survives for long periods in the open sea and during deep dives: sucking lice, Echinophthiriidae (Phthiraptera: Anoplura)[3,4]. They are obligate ectoparasites of pinnipeds living attached to their hosts' fur and feeding on their blood[5,6]. During their evolution, echinophthiriids had to adapt to a challenging new environment, when the ancestors of their recent host returned from land to sea in the Miocene[7–9]. Thereby, these ancestrally terrestrial insects adapted gradually to challenging physical conditions in extremely changing environments with high salinity, fluctuating temperatures, hypoxia and exceptionally high hydrostatic pressure[10]. The adaptation to this new way of life impacted morphology, reproduction, and distribution of the parasites[11,12].

One representative of Echinophthiriidae that has to face the challenges of surviving on its host under adverse conditions is the seal louse, *Echinophthirius horridus*. This insect parasitises true seals (Phocidae) and is most common on harbour seals (*Phoca vitulina*) and grey seals (*Halichoerus grypus*)[3,4,13], which occur in various insular and coastal habitats in the Northern Hemisphere[14,15]. During their dives, which can reach depths between 450–631 m and last 20–35 min[16–21], they are exposed to temperatures as low as 0 °C[22,23]. As part of their regular haul-outs ashore, they also have to withstand temperatures of up to 28.6 °C[23–25]. Therefore, in addition to these extreme temperature fluctuations and a hydrostatic pressure of about 60 kg*cm⁻² (5883.96 kPa) at 600 m depth[26], the seal lice living on the surface of these seals also need to stay firmly attached to the seals' fur despite a swimming speed of 18 km/h of their hosts[27].

Strong and reliable attachment of insect parasites on host hairs in general is a challenging task: proper contact formation of the attachment system to the substrate becomes more difficult due to complex surface topology and the flexibility of hairs[28]. Since the seal louse is a permanent and obligate ectoparasite, its survival is crucially dependent on permanent contact to its host[29]. For

example, they feed on the blood of seals[29], consequently a loss of contact by the parasite during dives would mean its certain death in open waters.

For all these reasons, attachment to seal fur requires special adaptations to remain on this challenging surface. Previous studies on other species of the Echinophthiriidae already described that the tibiotarsus of the meso- and metathoracic leg pairs are strongly adapted to clamping fibrillary substrates and the first pair of legs is suggested to contribute to sensory perception due to its smaller and slender appearance[10,30].

However, the detailed functional mechanism of this attachment system, how it works in comparison to other systems and how efficient its performance is, remained unexplored. Therefore, the aim of the present study was (i) to analyse the morphology and material composition of the attachment system of *E. horridus* by using scanning electron microscopy (SEM), confocal laser scanning microscopy (CLSM) and histological sectioning and staining[31,32] and (ii) to measure attachment forces generated by the seal lice, when attached to the seal fur underwater. The obtained results were compared to other claw-based attachment systems of parasitic and non-parasitic insects, in order to shed light on the processes aiding in strong, reliable and reversible attachment on complex structures in the deep sea. This study additionally offers inspirations for technical development of biologically-inspired underwater grippers.

## Methods

**Animals.** Adult seal lice (*Echinophthirius horridus*; Anoplura; Insecta) (adulthood determined according to Scherf[33]) were collected during necropsies of harbour seals (*Phoca vitulina*) and grey seals (*Halichoerus grypus*), which was performed at the "Institut für Terrestrische und Aquatische Wildtierforschung" (ITAW), Büsum, Germany. The seals were found freshly dead or moribund along the North and Baltic Sea coast of Schleswig Holstein, Germany between May and November 2022 (Fig. 1a–c; see Supplementary Data 1 for coordinates of collection points). All lice investigated in the context of this study originated from seals examined in the frame of monitoring programs within the

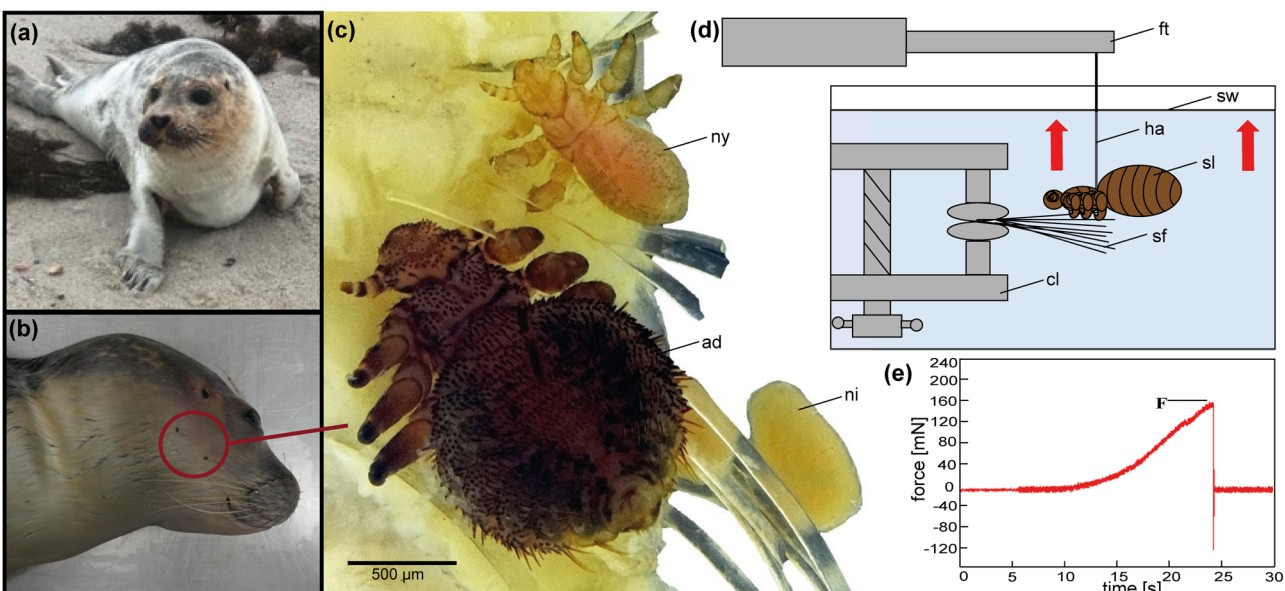

**Fig. 1 Focal species and experimental setup. a** The host of *E. horridus*, the harbour seal *P. vitulina*. **b** *Echinophthirius horridus* in the head region of *P. vitulina*. **c** Adult (ad), nymph (ny) and nit (ni) of *E. horridus* attached to seal fur. **d** Experimental setup. The seal louse (sl) was fixed with a human hair (ha) to a force transducer (ft) and actively pulled off from seal fur (sf), which was fixed with a clamp (cl) in an aquarium filled with sea water (sw). **e** Representative force-time curve showing the pull-off force at detachment (F). Figure 1a provided by Thomas Diedrichsen.

stranding network of Schleswig-Holstein to investigate their health status[34]. Seal lice were stored in a fridge at 4–8°C in single plastic beakers equipped with paper towels wettened in North or Baltic Sea water, respectively. Samples for morphological studies were stored in 70% ethanol. All host animals in our study were found dead, died naturally or were euthanized based on welfare grounds and none of the host animals was killed for the purpose of this study.

**Scanning electron microscopy (SEM).** Adult seal lice ($n = 5$) were dehydrated in an ascending alcohol series and subsequently critically point-dried in an automatic Leica EM CPD300 (Leica, Wetzlar, Germany). Afterwards, samples were sputter-coated with a 10 nm gold-palladium layer (Leica Bal-TEC SCD500). Specimens were scanned from both sides using a rotatable sample holder[35] at 15 kV acceleration voltage with a Hitachi TM3000 (Hitachi Ltd., Tokyo, Japan). Details of the tarsal morphology were examined using cryo-scanning electron microscopy (SEM) by freezing lice in a cryo stage preparation chamber at −140 °C (Gatan ALTO 2500 cryo preparation system, Gatan Inc., Abingdon, UK). Subsequently, frozen samples were sputter-coated with gold-palladium (thickness 6 nm) and observed with a cryo-SEM Hitachi S-4800 (Hitachi Ltd., Tokyo, Japan) in frozen condition at 3 kV accelerating voltage and −120 °C stage temperature. Obtained images were processed using Adobe Photoshop CS6 (Adobe Photoshop CS, San José, USA) and Affinity Photo 1.10.6 (Serif Ltd, Nottingham, UK).

**Confocal Laser Scanning Microscopy (CLSM).** For CLSM analysis, seal lice ($n = 5$) kept in 70% ethanol were hydrated in a descending alcohol series, transferred into glycerine ($\geq 99.5\%$) and mounted under a high precision cover slip (thickness = 0.170 mm ± 0.005 mm, refractive index = 1.52550 ± 0.00015, Carl Zeiss Microscopy GmbH, Jena, Germany) prior to scanning. The autofluorescence of the samples was analysed using the CLSM Zeiss LSM 700 equipped with the upright microscope Zeiss Axio Imager (Carl Zeiss Microscopy GmbH, Jena, Germany) and four stable solid-state lasers (wavelengths 405, 488, 555 and 639 nm and emission filters BP420–480, LP490, LP560, LP640 nm, corresponding to detected emission wavelengths 420–480 nm visualised in blue, $\geq 490$ nm in green, and $\geq 560$ nm and $\geq 640$ nm in red). Following Michels & Gorb[32], the 405 nm laser in combination with the bandpass emission filter transmitting 420–480 nm was used to visualise less sclerotised cuticle potentially containing high proportions of resilin. To detect more sclerotised cuticle, lasers with wavelengths of 488 and 555 nm in combination with long-pass emission filters of 490 and 560 nm were used. In addition, the 639 nm laser in combination with the 640 nm long-pass emission filter was applied to visualise autofluorescence beyond this range[36]. The obtained autofluorescence signals were transferred into maximum intensity projections using the ZEN 2008 software (www.zeiss.de/mikroskopie) and subsequently processed in Adobe Photoshop CS6 (Adobe Photoshop CS, San José, USA). Thereby, the received projections allow for a qualitative description of the material composition of the cuticle, but do not represent a precise quantitative measurement of the cuticle stiffness[32,36–39]. Different colours correspond to specific autofluorescences[32]: (1) reddish autofluorescence represents highly sclerotised cuticle, whereby the higher the red content, the higher the degree of sclerotisation; (2) greenish autofluorescence corresponds to relatively tough cuticle; (3) and bluish autofluorescence represents soft, less-sclerotised cuticle.

**Force Measurements.** In order to evaluate the maximum attachment force of the seal lice on the seal fur, force

measurements were performed (Fig. 1d, e). To measure the pull-off force required to detach a louse from the seal hairs, a BIOPAC MP 100 data acquisition system (BIOPAC System Inc, Goleta, USA) was equipped with a Fort100 force transducer (100 g capacity, World Precision Instruments Inc., Sarasota, USA) (Fig. 1d). The latter was fixed to a compact linear stage with a stepper motor (Physik Instrumente GmbH & Co. KG, Karlsruhe, Germany) to enable targeted computer-controlled movement of the force transducer. A human hair was connected to the force transducer and on the other end tied up to the base of the abdomen of the seal louse, to enable active pulling-off the louse from the seal fur in vertical direction in the sense of a pulling thread. An aquarium filled with Baltic or North Sea water depending on the origin of the seal lice was equipped with a clamp to fix cut-off seal fur underwater (Fig. 1d). Subsequently, the seal lice fixed with the human hair were brought into contact with the seal fur (as attachment substrate) underwater until the specimens held tight on to the seal fur. Tension was applied to the human hair by moving the force transducer in vertical direction until the seal lice were detached from the seal fur (Supplementary Video 1). By using the software AcqKnowledge 3.7.0 (BIOPAC System Inc, Goleta, USA), force-time curves were recorded and the maximum attachment forces were determined (Fig. 1e). The weight of the individual lice was measured with a Sartorius ultra-microbalance MSE 2.7S-DM (Sartorius, Göttingen, Germany; ≤ 0.00025 mg). To account for the different weights of different lice individuals, the safety factor (SF) based on the relationship between maximum attachment force ($F_a$) and the weight force ($F_w$) was calculated:

$$SF = \frac{Fa}{Fw}$$

The weight force ($F_w$) was calculated based on the gravitational constant ($g$) and the mass of the individual louse ($m$):

$$Fw = m * g$$

A total of 21 seal lice were each measured three times daily for at least three consecutive days to test the progression of the detachment forces in repetitive cycles. Thereby, the number of measured individuals declined in the course of the experimental days, as the individuals died successively likely due to lack of food. The measured attachment forces are listed in the Supplementary Data 2.

**Histological sectioning and staining.** During necropsies, lice were removed carefully with a forceps or louse comb off the seal ($n = 3$). They were then cleaned with water, collected and fixed in 5% glutaraldehyde. To increase the permeability the lice were incubated for 1 h in 0.1% saponine in cacodylate buffer. Subsequently, they were pre-contrasted in 1% osmium acid in cacodylate buffer and then dehydrated and embedded in epoxid resin according to a standard laboratory procedure. Semithin sections of 1 μm thickness were cut using a rotation microtome 2030 (Leica, Wetzlar, Germany). Standard Toluidine Blue staining protocol was used for staining the sections[40]. Thereby, Toluidine Blue stains basophilic tissue components and has been shown to stain less-sclerotised cuticle in sapphire to bright blue colouration, while solid cuticle is not stained at all[41]. Stained sections were studied in a binocular light microscope Olympus BX 53 (Olympus Europa SE & Co. KG, Hamburg, Germany).

**Statistics and reproducibility.** We compared the daily performance of all lice based on mean values of the three daily measurements for every louse per test day using a Kruskal-Wallis one-way analysis of variance (ANOVA) on ranks and a subsequent Dunn's post hoc test with a significance level of 0.05 as

the data was not normally distributed (criterion for normal distribution: Shapiro Wilk test, $p > 0.05$).

The overall maximum attachment forces of all lice were calculated by selecting the highest average value across the different measurement days per louse. To test for differences in the attachment safety factors between the sexes, we performed a Mann-Whitney-U-test as the data was not parametric (criterion for normal distribution: Shapiro Wilk test, $p > 0.05$). Moreover, we tested a linear regression on calculated mean safety factors and weight of the individuals to analyse whether size might have an impact on attachment performance. All statistical analyses were performed in R (version 4.2.1, the R Core Team 2022), except for the retro perspective power analysis, which was performed in SigmaPlot 12.0 (Systat Software Inc., San José, USA). R scripts can be found attached in Supplementary Data 4.

**Reporting summary**. Further information on research design is available in the Nature Portfolio Reporting Summary linked to this article.

## Results

**Morphology of the lice attachment system**. *Echinophthirius horridus* has three pairs of scansorial legs, which are similar in length and end in strong acuminate claws (Fig. 2a–d). The legs are composed of coxa (cx), trochanter (tr), femur (fe), and the segment consisting of tibia and tarsus is fused to one tibiotarsus-complex (tbta) ending in a pretarsal claw (tcl) terminated by powerful grasping organs for efficient attachment to the host hair. The tibiotarsus is composed of a single curved claw with blade-

like ridges (bd) in the distal half on the inner side and an euplantula (ep) covering the proximal half of the claw (Fig. 2e–f). The counterpart is a thumb-like elongation (thu) of the tibiotarsus-complex from which four blade-like sculptured setae (se) protrude. Furthermore, a tibial pad (tp) with many sensory sensilla on its surface and elongated spiny setae (se) can be found on the claw pendant.

When considering the material composition of the cuticle it becomes obvious that the tibiotarsus-complex appears to be highly sclerotised. The only exceptions are the euplantula and the tibial pad, which are dominated by blue autofluorescence and therefore, presumably, less sclerotised and soft (Fig. 2b–e). The four blade-like sculptured setae on the tip of the thumb-like claw counterpart (thu) appear highly sclerotised, while the elongated spiny setae show a greenish autofluorescence signal and are therefore less sclerotised (Fig. 2b–d, f).

In addition, it is noticeable that the cuticle of the leg is always more sclerotised on the proximal part of each leg segment next to the joints, while less sclerotised regions can be found on the distal side of each leg segment overlapping the proximal side of the subsequent segment (Fig. 2b–d, h).

**Analysis of the attachment posture of *E. horridus* on seal fur**. Attachment sites of adult seal lice on individual seal hairs were analysed. In general, lice orient their head region during interlocking in the direction of the hair root of the seal, so that the water current flows from their head over the abdomen during diving (Fig. 3a).

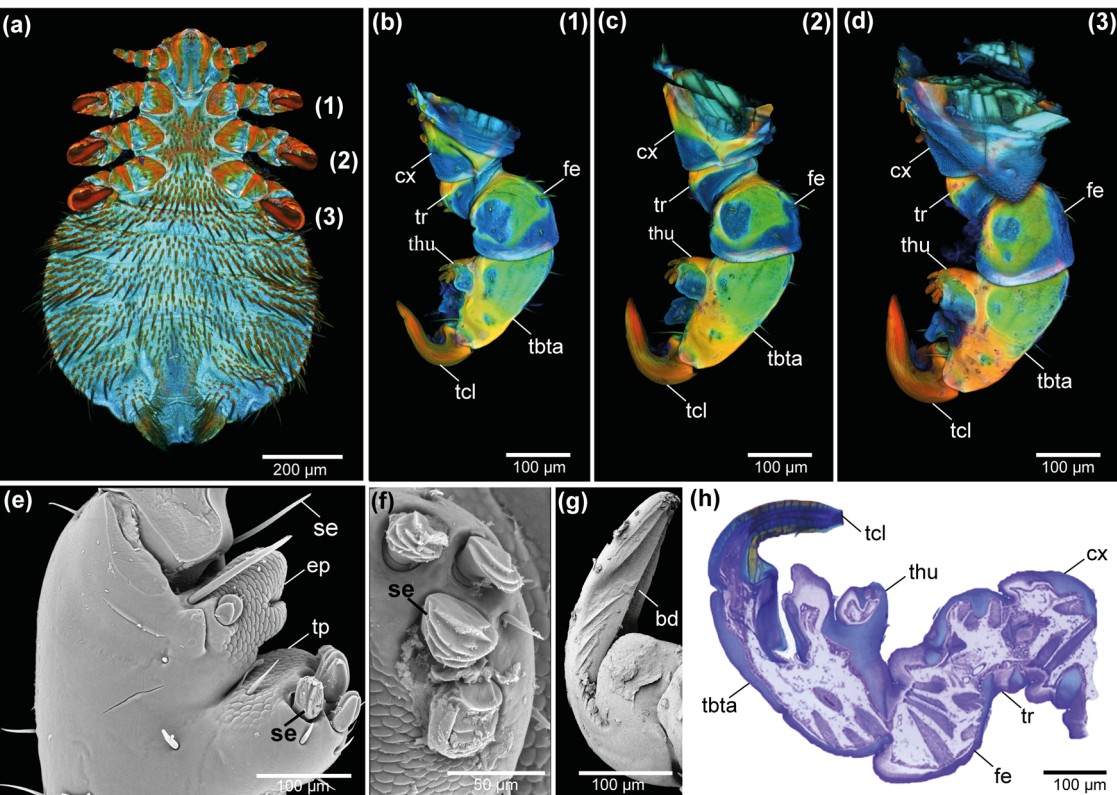

**Fig. 2 Morphology of the attachment structures of *E. horridus*.** Confocal laser scanning microscopy maximum intensity projections of **a** the entire specimen from ventral view, **b** the first leg from ventral view, **c** the second leg from ventral view, and **d** the third leg from ventral view. Scanning electron microscopy images of **e** euplantula and tibial pad on the tibiotarsus-complex, **f** the four blade-like setae on the thumb-like counterpart of the claw, and **g** the blade-like structures on the inner side of the claw. **h** Histologically sectioned and stained slide of the second leg. bd blade-like ridges on the inner side of the claw, cx coxa, ep euplantula, fe femur, se setae, tbta tibiotarsus-complex, tcl tarsal claw, thu thumb-like counterpart of the tarsal claw, tp tibial pad, tr trochanter.

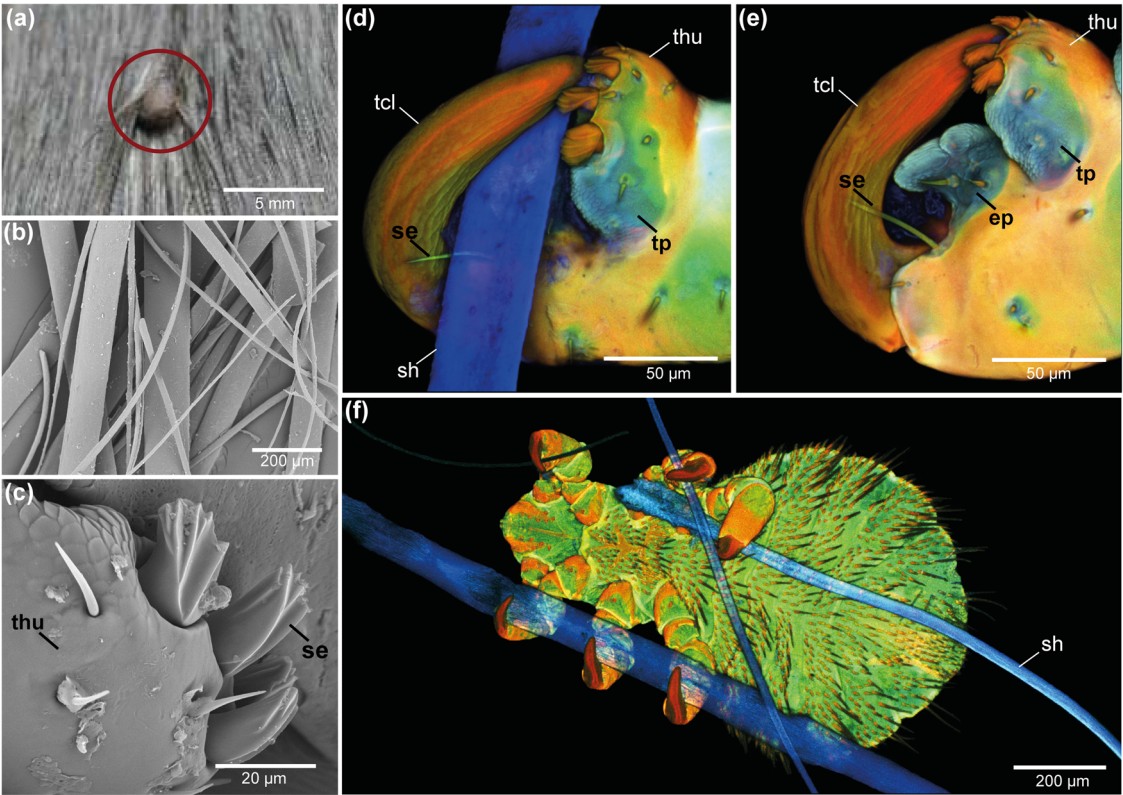

**Fig. 3 Attachment of *E. horridus* on the seal fur.** Scanning electron microscopy image of **a** the seal louse orientation on seal fur, **b** individual hairs of seal fur, and **c** blade-like segmented setae on the thumb-like counterpart of the claw. Confocal laser scanning microscopy maximum intensity projections of **d** the tarsal claw clinging to the seal hair, **e** the tarsal claw closed without seal hair, and **f** the seal louse clinging to seal hairs with all six legs. ep euplantula, se setae, sh seal hair, tcl tarsal claw, thu thumb-like counterpart of the tarsal claw, tp tibial pad.

During attachment, louse legs are bent and positioned in such a way that the louse body is in close contact to the seal fur. If possible, the louse grabs on the hair with all six claws to maximise the number of clamping points (Fig. 3f). When the claw is in contact with its thumb-like counterpart without attaching to seal fur, the euplantula and the tibial pad are in close contact with each other (Fig. 3e). However, if the louse clings to a hair (sh), the euplantula and the tibial pad are in direct contact with the hair (Fig. 3d). The short blade-like setae on the tip of the claw pendant touch both the hair and the tip of the claw, while the elongated spiny setae are tightly pressed against the lateral side of the claw (Fig. 3c, d). Furthermore, the seal hairs themselves have a flattened cross-sectional shape with a width of 10-150 μm. The seal hairs have their elasticity modulus in the range of 1-4 GPa[42] and therefore blade-like ridges on the inner side of the lice claw can easily press into the keratinous material of the hair (Fig. 3b, d, f). A video showing the movement of *E. horridus* on seal fur and the clamping of claws on the seal hairs is provided in the Supplementary Video 2.

**Force measurements of *E. horridus* on seal fur.** Force measurements with adult seal lice were performed on cut-off seal fur, fixed with a clamp underwater. After proper contact formation, the seal lice were actively pulled off from the seal fur. The maximum measured attachment force averaged over three measurements per day was 49.65 ± 20.16 mN and the corresponding safety factor was 3627.19 ± 1146.88 (median ± mean deviation from the median). Thereby, the safety factor represents how many own body weights the seal louse could hold attaching to the seal fur (attachment force divided by weight force). In individual measurements, some lice even reached maximum attachment forces up to safety factors of 18000 (Supplementary Data 2).

No statistically significant differences could be found between both sexes (Mann-Whitney-U-test, $p = 0.7856$) and the lice showed a tendency for higher attachment forces during the first three days of measurements, however, the measured values were not significant either (Kruskal-Wallis-ANOVA, $p = 0.016$; Dunn's post hoc test, all $p > 0.35$), although a retro perspective power analysis on the measured attachment forces revealed sufficient expressive sample size (Fig. 4).

Since body weight scales with the linear body dimensions with the exponent of 3 (similar to the volume) and force with the exponent of 2 (due the force dependence on the cross-section of muscles), a logarithmic dependence with a slope of 0.89 between measured attachment force and weight was observed ($p = 0.0045$; $R^2 = 0.3535$).

## Discussion

Seal lice show exceptional attachment strength on seal fur and are able to attach repeatedly and reliably. This ability is especially necessary, because they have to cope with great challenges concerning their habitat, such as high hydrostatic pressure at diving depths of ca. 600 m[19], hypoxia due to dives of 20–35 min[16], fluctuating temperatures[22,24], and extreme drag forces at swimming speeds of seals up to 18 km/h[27]. However, since they are obligate ectoparasites of marine mammals and dependent on the blood of their hosts[5,6], adaptations concerning their morphology, reproduction and distribution are necessary to ensure their survival over generations[11]. One of the solutions to these challenges is an extraordinary attachment system, which allows both strong clamping, and repeated release of the clamping to ensure mobility of the parasites for new host recruitment during haul-outs.

Based on our findings, the claws of the lice are the main device for this strong and reliable attachment (Fig. 2; 3c–f). When

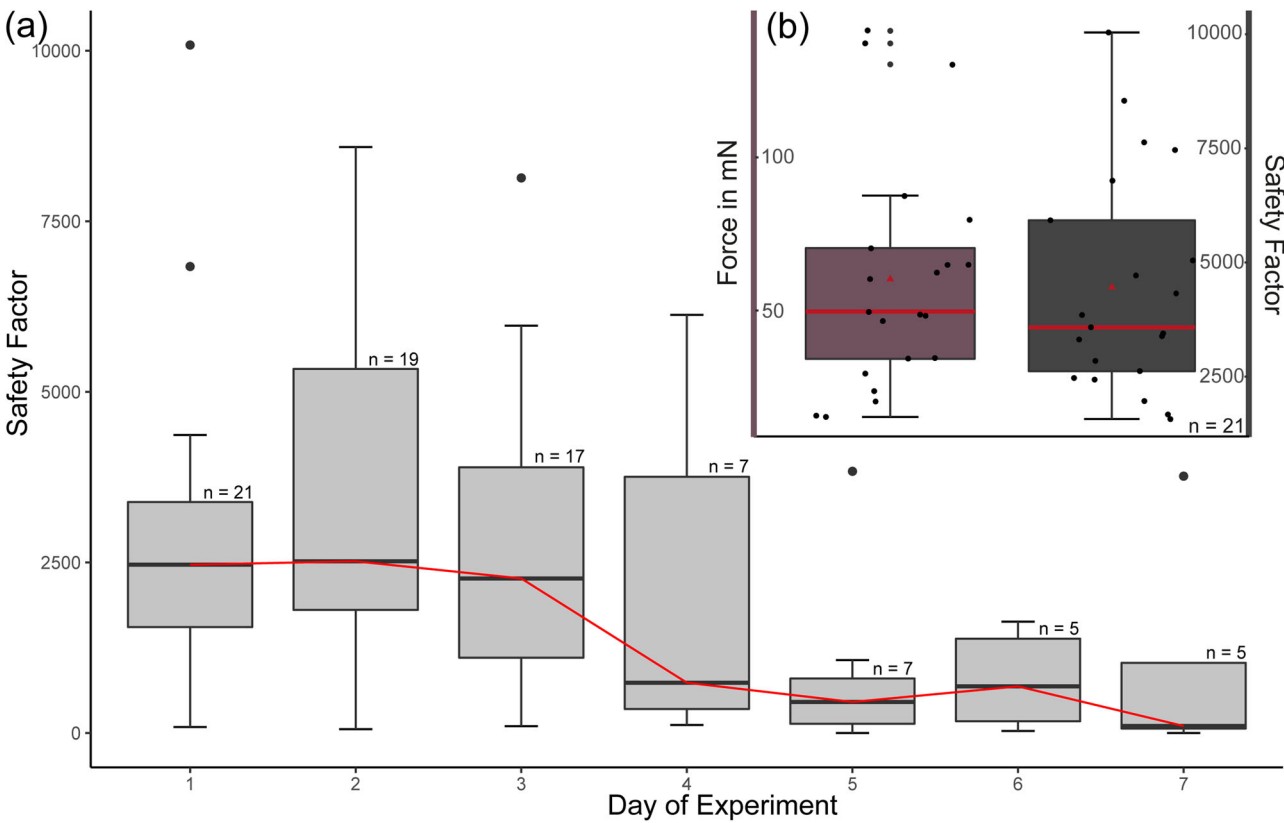

**Fig. 4 Attachment forces of *E. horridus* on the seal fur. a** Boxplots showing the daily performance displayed as safety factor (pull-off force divided by the weight force) of all measured individuals within 7 days of experiment. Values are based on the mean of three measurements per day per individual, respectively. **b** Boxplots showing the attachment force and calculated safety factors prior to the louse detachment from seal fur. In total, 21 individuals were measured and the mean of the three measurements of the day with maximum attachment force was used for each individual. The boxes indicate 25th and 75th percentiles, the line within the boxes represents the median, and whiskers (error bars) define the 10th and 90th percentiles.

comparing Phthiraptera with its sister-group Liposcelididae[43–45], differences in the claw structures are noticeable as a result of the specialised parasitic lifestyle of Phthiraptera. While Liposcelididae possess two serrate claws and tarsomeres densely covered with microtrichia, Phthiraptera have only one large claw including a thumb-like counterpart and a pad-like euplantula on the ventral side of the tarsus[46]. As indicated by the CLSM results, *E. horridus* possesses a second smooth pad consisting of the soft cuticle, which we called the tibial pad. The euplantulae of Phthiraptera likely evolved convergently to the euplantulae of other insects[46,47]. Moreover, even within the Anoplura, there are differences in leg and claw structures: for example, in the legs of the hog louse (*Haematopinus suis*), a clear distinction between the tibia and tarsus is possible, whereas these segments are fused in the seal louse (Fig. 2b–d). Furthermore, the thumb-like counterpart of the claw is most highly developed in the seal louse, while in other sucking lice, like the human head louse (*Pediculus humanus capitis*), this structure consists only of two spines[48]. Moreover, the claw itself is distinctly broader and more brute than is the case of all other species within Anoplura[49].

The terminology of the soft cuticular pads involved in the attachment mechanism in sucking lice has been handled differently by previous authors. While some authors called the central pad euplantula[46,50,51], other used empodium for it and pulvillus for the smaller pad on the tibial process[52]. The terms empodium and pulvillus, however, do not apply to the structures found here, as empodia are defined as processes of the unguitractor plate and pulvilli as paired lateral lobes located on the auxiliae[51]. Therefore, the proper term for the ventral soft cuticular pad on the tarsal part of the tibiotarsus-complex is, by definition, euplantula[51]. The

proximal pad on the distal tip of the tibial portion of the tibiotarsus-complex, however, is in contrast likely not part of the original tarsus, but more similar to the accessory attachment pads on the tibiae of some insects[51]. Such pads are convergently evolved in different lineages of Heteroptera and called *fossula spongiosa* in many cases[47], and in Phasmatodea in form of an accessory euplantula on the tibial tip[53]. These structures clearly evolved convergently[47] and we, hence, call the smooth cuticular pad on the tibial process herein tibial pad to emphasize the independent origin of this structure.

When comparing the attachment structures of the seal louse with those of other ectoparasitic insect species, remarkable structural differences become obvious. For example, the louse fly *Crataerina pallida* attaches to its bird host *Apus apus* by piercing its tridentate claws into the feather vanes making single claw barbs hook between clamping regions[54]. A similar system has been previously found in the ectoparasitic bee louse *Braula coeca*, which attaches to its host *Apis mellifera* by using 24 clamping regions with claw teeth interlocking with the bee hairs[28]. However, this kind of attachment is not transferable to the seal louse since feather microstructures as well as bee hairs have a much smaller diameter than seal hair, so that this type of comb structure would not work for seal lice. Instead, the claw of the seal louse is strongly adapted to the attachment on the flattened hair of the seal by clamping various diameters of seal hair (10–150 μm) between the euplantula of the claw and the tibial pad of the tibiotarsus-complex and the thumb-like counter structure equipped with spines (Fig. 3). Parasites attaching to comparable hair structures also belong to the Hippoboscoidea like *C. pallida*, but adhere to mammalian fur by using an attachment

system consisting of two claws and two pulvilli. However, they do not use a lice-like closure system based on a claw and a counterpart, but they merely seize the hairs within the claw gaps or/ and adhere to thicker hairs with their adhesive pads (pulvilli)[55]. As a consequence, the seal louse can perfectly attach to its host's hairs and its way of attachment also works on different similar substrates with comparable diameters as it has already been shown for feather lice[56]. Nevertheless, the seal louse is still dependent on their specific hosts for reproduction and nutrition[29].

In contrast to the above-mentioned representatives of different ectoparasitic insect groups, seal lice use a highly modified snap hook system for reliable attachment to seal fur, which, despite this very strong clamping performance, also facilitates detachment. The system based on an euplantula and tibial pad pressed against the clamped hair from opposite sides when the claw is closed has already been described in other terrestrial lice species and most likely serves to increase friction during clamping[52]. This assumption is supported by our CLSM results, demonstrating for the first time, to the best of our knowledge, that these two pad structures appear very soft and can presumably adapt stronger to the flattened form of the seal hair, while the blade-like ridges on the inner side of the claw are pressed into the soft material of the hair (Fig. 3d, e). We assume that these ridges help to squeeze the seal hairs into the base of the louse claw and thereby increasing anisotropic friction[57–59].

Usually, insects use the tip of their claws for reliable mechanical interlocking and friction increase with rough substrates[60–65], whereby the claw tips show high stiffness and high material strength[63,66]. In *E. horridus*, the whole claw and even the distal half of the tibiotarsus-complex appears to be highly sclerotised (Fig. 2a–d; 3d, e) indicating possible presence of high stress concentrations acting on these structures during interlocking[28,66]. This may also offer an explanation for why the tibia and tarsus are fused in the seal louse, although this is uncommon even within Anoplura[49]. The proximal part of each leg segment next to the joints appears more sclerotised, while the more distal part of each leg segment shows less sclerotisation. This might be due to the possibility that these cuticular parts might contain the elastomeric protein resilin for higher resilience, lower fatigue, and stronger damping[67] (Fig. 2b–d, h).

While in head lice, for example, the claw tips protrude beyond the opposite spines, when closed[68], the tip of the claws of seal lice meets four short, flattened setae with fine groove structures on their surface (Fig. 2e, f; 3c–e). We assume that these highly modified setae with small resilin rings on their sockets for higher flexibility are used as stopper mechanism to avoid lateral deflection of the claw and enable stabilisation of the claw-tibiotarsus-complex during interlocking. Furthermore, we suppose the lateral prolonged setae consisting of less sclerotised material to have a sensory role to provide the louse with the information whether or not a host hair is properly clamped.

In the context of its attachment performance the seal lice showed exceptionally high attachment forces on the seal hair with average safety factors of 4500 and individual maximum values of 18000 (Fig. 4; Supplementary Data 2). To put these attachment forces into context with other insects using their interlocking systems, we compared them to a selection of maximum attachment values of different insect groups mentioned in the literature (Fig. 5; Supplementary Data 3). However, a direct comparison between these values is limited, since the safety factors have been obtained with different experimental methods from various surfaces and none of these other species used is aquatic. Nevertheless, this comparison is helpful in our opinion for the general understanding of performance of tarsal interlocking devices in insects. When comparing the measured attachment forces, it is noticeable that safety factors of parasitic and non-parasitic insects differ significantly. While non-parasitic insects showed maximum safety factors around 350 (Formicidae; *Atta cephalotes*)[64], the avian ectoparasite *C. pallida* and the bee ectoparasite *B. coeca* generated safety factors of around 1000–3000[28,54]. Thereby, the first five presented insect species use their claws as single-tip-interlocking systems for attachment to the substrate[62,64,69,70], where the claw tip interacts with surface asperities. *Crataerina pallida* relies on a tridentate claw system that clamps into the fibrous substrate[54] and *B. coeca* even uses a comb-like structure to properly attach to the bee hair by clamping a relatively higher number of fibrous elements with the multiple claw tips[28] (schematically shown on the x-axis of Fig. 5). However, these values are exceeded by far by the seal louse, which reached safety factors of about 10000. In water insects, the highest attachment performance ever measured was recorded for Blephariceridae larvae using specialised suction cups reaching safety factors of about 320 to 1120[71]. Hence, to our knowledge, the seal louse therefore generated the highest safety factors ever measured before for any insect.

For parasites strong and reliable attachment is vital, since they are dependent in various ways of their hosts and thereby experiencing a high evolutionary pressure to stay in direct contact with their hosts[72]. For this reason, the morphological adaptations of the attachment system, as well as the resulting extraordinarily high attachment forces are well explainable. The special feature of these attachment systems, however, is that it also allows for easy detachment, which is not the case in other organisms with comparably high safety factors that attach permanently to a surface (e.g., barnacles, mussels, and tubeworms)[73]. Strong, reliable and reversible attachment is essential for seal lice, since they must be able to look for new host animals during haul-outs for distribution and reproduction, but also to make sure that they do not loose contact to their hosts during deep dives. Their marine habitat imposes very special demands on the seal lice like extraordinarily high Reynold's numbers due to the swimming speed of seals and the high fluid density of water. Therefore, higher safety factors of attachment might be important for marine parasites compared to terrestrial parasites. To get an idea of the forces acting on the lice when diving on the seals, we calculated the drag force, a single louse is exposed to, on the most uncovered area on a seal by using the following formula[28]:

$$D = \frac{C_d * p * S * v^2}{2}$$

$C_d$ is the drag coefficient of a sphere resulting from the calculated Reynold's number of a seal louse in water, $p$ is the fluid density of water of 1000 kg*m$^{-3}$, $S$ is the area resisting the water flow, and $v$ represents the swimming speed of the seal of 4.9 m*s$^{-1}$ [27] (Supplementary Data 5). The resulting drag force of 0.03394 mN put in relation to the much higher attachment force of the seal louse of 60.23 mN, reveals that the louse is approximately 1775 times stronger than the experienced drag force. Although this is a very simplified calculation, we can assume that the seal louse finds sufficient attachment on the seal during deep dives at full speed. Furthermore, the lice usually sit in the middle of the seal fur completely covered by seal hair next to the skin of the seals and thereby being less exposed to flow and finding shelter. Accordingly, the attachment structures of the seal lice are probably even a kind of overdesign in comparison to the experienced drag forces.

Furthermore, we found a significant correlation between weight and measured attachment forces of seal lice individuals. Thereby, the slope of the logarithmic trend line is hyper allometric, which suggests that weight might be less important during attachment than expected under the assumption of a slope of 2/3 for logarithmic data. This gives first indications that a passive

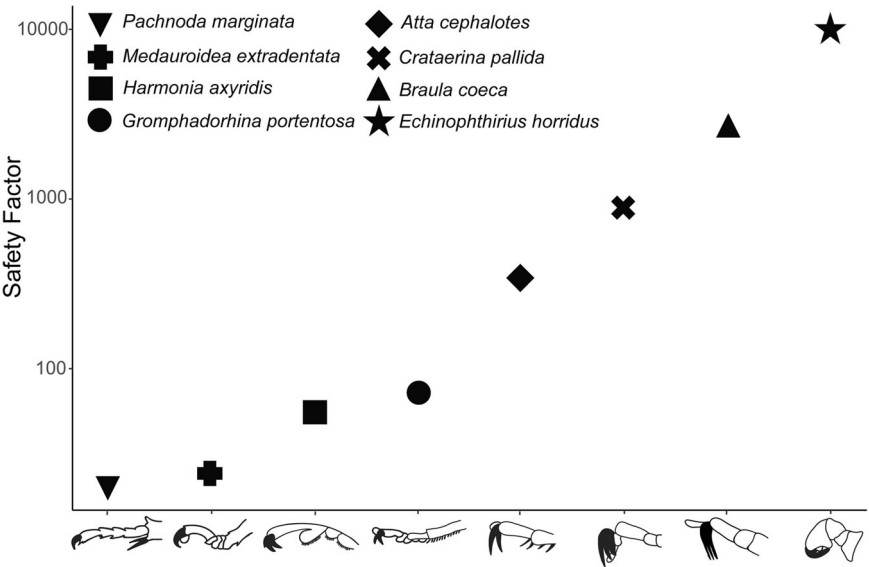

**Fig. 5 Safety factors of *E. horridus* (this study) and other insects using interlocking systems for attachment (literature data).** Y-axis is a logarithmic scale. Structures of the legs involved in interlocking are marked in black on the x-axis in the schematic representations of the different tarsus types. A detailed list of all reported maximum safety factors can be found in the supplementary materials (Supplementary Data 3).

form of interlocking, which can be found in *C. pallida* for example[54], and also an active use of musculature might play a role during their attachment. This assumption is also strengthened by the fact that we were able to repeatedly measure large values in the individuals on different days (Supplementary Data 2). From this result, it can be concluded that the clamping is released, but the system is not (remarkably) damaged. Otherwise, the attachment force would be high only once. Potentially, this could mean that the musculature keeps the claw closed and that the claw is not broken. This hypothesis is also supported by the observation that the measured attachment forces decline with proceeding experimental days (Fig. 4), since the animals are probably increasingly exhausted, due to the lack of food and therefore have less energy available for muscle contraction. Conversely, however, this also illustrates once again how important direct contact of the parasite with the host is in this case and how strongly the louse depends on the seal.

In conclusion, the seal louse, *E. horridus*, shows various morphological adaptations for attachment underwater on the seal fur: soft pad-like structures (euplantula and tibial pad) for higher friction, when attached to the seal hair, sensory setae for assuring proper interlocking of the claws including stopper-mechanisms, a fused tibiotarsus-complex for higher stability during attachment, and blade-like structures on the inner side of the claw piercing into the seal hair for better grip on the soft material. Based on comparisons with other ectoparasites and even closely related species within Anoplura, it is therefore reasonable to speculate that the attachment structures of the seal louse are special adaptations to life in a highly dynamic marine environment.

We assume that these morphological adaptations are responsible for the extraordinarily high attachment forces of seal lice on seal fur. On average, the lice achieved safety factors (attachment force divided by weight force) of about 4500 and even 18000 as the absolute highest measured value of the whole dataset. Thereby, to our knowledge, they show the highest attachment forces ever measured for insects.

Based on our results, it would be interesting for future studies to take a closer look at the musculature involved in the attachment of marine lice, to reveal the mechanisms underlying these forces. We expect that the adaptations to marine life required changes in the morphology of parasitic organisms for survival in extreme environments. This kind of research additionally offers inspirations for technical development of biologically-inspired underwater grippers.

## Data availability

All data is provided in the supplementary material of the manuscript.

## Code availability

All scripts used in this manuscript for graphs and analyses are provided in the supplementary material of the manuscript.

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

## Acknowledgements

We would like to thank Helen Gorges, Fabian Bäumler, Julian Thomas, and Benedikt Josten for advice and support during this study. Furthermore, we thank Esther Appel, Dr Alexander Kovalev and Kerstin Rohn for technical support during the experiments and Prof. Dr Olivia Roth for the provision of Baltic and North Sea water. We are grateful for the Stranding Network Schleswig-Holstein, and in particular for the seal rangers Sönke Lorenzen, Thomas Diedrichsen and Rolf Lorenzen for alerting us at seal finds and RDC Autozug Sylt with Sandy Kruse for logistical support. Funding to S.N.G. by the grant GO 995 46-1 from German Science Foundation (DFG) within the Special Priority Program (SPP 2332) "Physics of Parasitism". The funders took not part in the study design, data collection and analysis, decision of publishing or any preparation of the manuscript.

## Author contributions

A.P., K.L., S.N.G.—Conceptualisation; A.P., S.N.G.—Data curation; A.P., T.H.B., S.N.G.—Formal analysis; K.L., S.N.G.—Funding acquisition; A.P., T.H.B., I.H.—Investigation; A.P., T.H.B., P.W., S.N.G., I.H.—Methodology; KL, SNG - Project administration; K.L., S.N.G.—Resources; A.P., T.H.B.—Software; K.L., S.N.G.—Supervision; A.P., K.L., S.N.G.—Validation; A.P.—Visualisation; A.P.—Writing—original draft; I.H., T.H.B., P.W., K.L., S.N.G.—Writing – review & editing.

## Funding

## Competing interests

All authors declare no competing interests.
