## [Peer Review File · Communications Biology]

Reviewers' comments:

Reviewer #1 (Remarks to the Author):

Dear authors,

Overall the text has a very clear structure and lacks excessive or confusing information. The introduction provides the reader with a good background on the sucking lice. The technical part of the project is well documented and the discussion provides an excellent explanation of the actual meaning of the experiment results.

Please find below some suggestions to improve the manuscript (with line numbers):

25: Please change "strongest insect attachment forces" to "strongest relative insect attachment forces" (larger insects could have higher absolute forces).

35: The statement about the successfulness of insects is subjective, since it lacks a clear criterion. Any group that is more inclusive (Arthropoda, Metazoa, etc.) has a higher species richness and a larger number of inhabited habitats. I would suggest to rephrase this sentence.

51: Please use dashes instead of hyphens for number ranges throughout the whole manuscript.

125: Please consider adding a more precise description how the false color image was composed. Which bandwidths translate to the RGB colour channels? (reproducibility)

193: Since the statistical analysis is entirely independent from the graphical user interface, R Studio does not need to be cited.

Apart from those minor corrections and suggestions, I highly recommend this manuscript for publication and I think that it is a very sound and valuable contribution to the study of insect biomechanics.

Reviewer #2 (Remarks to the Author):

The authors present, for the first time, an analysis of the features associated with the attachment of a seal louse to its host hair. Considering the great adaptive pressures on seals during the colonisation of the ocean by their pinniped hosts, this result is a key process. To understand the structure and how it works, they combine different techniques such as CLSM, SEM, and histology. The results show that the lice achieve an extraordinary attachment force. In this sense, the article could be a major contribution to understanding the adaptations of this particular group of insects. It is undoubtedly a valuable contribution, analysing a key process for this particular group of animals. I especially want to recognize the great work done with the figures. They are very clear and illustrative.

My main concern with this study relates to the fact that human hair was used for the experiment instead of seal hair. In the discussion, the author refers to the attachment to a seal capillary fibre. I think this is something that should be discussed. As the authors point out, seal lice have specific adaptations, many of which are very specific. So it would be possible that the diameter of the fibre is precisely related to the size or closure system of the claws. I suggest that this point be discussed.

Major comments:

- I found the title "ectoparasitic seal louse" redundant. By definition, lice are ectoparasites. The first sentence of the abstract postulates a co-evolutionary arms race between marine mammals and their parasites. However, this is not true for all types of parasites in these hosts. Moreover, there are no

insect parasites in cetaceans or sirenians. It would be easier to follow if the paragraph began with a reference to pinnipeds.

- When the results of pressure measurements are presented, the value does not have a unit, making it difficult to interpret the meaning.
- The main statement in the introduction suggests that lice are host-dependent because they feed on blood. However, lice are permanent and obligate ectoparasites, regardless of whether they feed on blood or dead cells, feathers, etc. I think this is actually the fundamental point about lice dependence on the host: they cannot survive outside of it. For seal lice, this is more extreme because they infest amphibious hosts.
- In Material and Methods it would be useful to provide the number of lice collected and analyzed under each method.
- The discussion should include at least one paragraph analysing what this means for lice. I suggest discussing Bush, S. E., Sohn, E., & Clayton, D. H. (2006). Ecomorphology of parasite attachment: experiments with feather lice. *Journal of Parasitology*, 92(1), 25-31.

Reviewer #1 (Remarks to the Author):

Dear authors,

Overall the text has a very clear structure and lacks excessive or confusing information. The introduction provides the reader with a good background on the sucking lice. The technical part of the project is well documented and the discussion provides an excellent explanation of the actual meaning of the experiment results.

Please find below some suggestions to improve the manuscript (with line numbers):

Reviewer comments	Response	Line
Please change “strongest insect attachment forces” to “strongest relative insect attachment forces” (larger insects could have higher absolute forces.	Response: We have changed “strongest insect attachment forces” to “strongest relative attachment forces”.	25
The statement about the successfulness of insects is subjective, since it lacks a clear criterion. Any group that is more inclusive (Arthropoda, Metazoa, etc.) has a higher species richness and a larger number of inhabited habitats. I would suggest to rephrase this sentence.	Response: We have changed “animal group” to “animal class” since the reviewer is correct in remarking that the term “group” might be misleading regarding phyla like Arthropoda etc.	35
Please use dashes instead of hyphens for number ranges throughout the whole manuscript.	Response: We exchanged hyphens by dashes for number ranges throughout the whole manuscript.	51, 88, 204, 209, 216, 218, 222, 294, 302, 312, 365, 371, 374
Please consider adding a more precise description how the false color image was composed. Which bandwidths translate to the RGB colour channels? (reproducibility)	Response: We added the detected emission wavelengths and corresponding colours to make our images more reproducible.	117 ff.
Since the statistical analysis is entirely independent from the graphical user interface, R Studio does not need to be cited.	Response: The reviewer is absolutely right in remarking that R studio is actually the user interface and that we should just mention the software R. We have changed this in the manuscript.	197

Apart from those minor corrections and suggestions, I highly recommend this manuscript for publication and I think that it is a very sound and valuable contribution to the study of insect biomechanics.	Response: Thank you very much, we are very pleased about the positive feedback by the reviewer.	
--	---	--

Reviewer #2 (Remarks to the Author):

The authors present, for the first time, an analysis of the features associated with the attachment of a seal louse to its host hair. Considering the great adaptive pressures on seals during the colonisation of the ocean by their pinniped hosts, this result is a key process. To understand the structure and how it works, they combine different techniques such as CLSM, SEM, and histology. The results show that the lice achieve an extraordinary attachment force. In this sense, the article could be a major contribution to understanding the adaptations of this particular group of insects. It is undoubtedly a valuable contribution, analysing a key process for this particular group of animals. I especially want to recognize the great work done with the figures. They are very clear and illustrative.

Reviewer comments	Response	Line
My main concern with this study relates to the fact that human hair was used for the experiment instead of seal hair. In the discussion, the author refers to the attachment to a seal capillary fibre. I think this is something that should be discussed. As the authors point out, seal lice have specific adaptations, many of which are very specific. So it would be possible that the diameter of the fibre is precisely related to the size or closure system of the claws. I suggest that this point be discussed.	Response: For our force measurements we used seal hair (fur) as attachment substrate for the lice. We only tied the lice with a human hair to the force transducer but did not use human hair as an attachment substrate, as we were aware that this would not have been the actual natural attachment situation for the lice. Thereby, the lice were offered a whole bunch of seal hairs so they were free to decide to which hair diameter they preferred to attach. To make this clearer, we added few explanatory sentences to the text.	134 ff.
I found the title "ectoparasitic seal louse" redundant. By definition, lice are ectoparasites. The first sentence of the abstract postulates a co-evolutionary arms race between marine mammals and their parasites. However, this is not true for all types of parasites in these hosts. Moreover, there are no insect parasites in cetaceans or sirenians. It would be easier to follow if the paragraph began with a reference to pinnipeds.	Response: We partially agree but the vernacular term "louse" is not exclusively used for sucking lice (Phthiraptera) as it is also commonly used for plant lice, book lice etc. which are not necessarily ectoparasites. Therefore, we would like to keep this title. To account for exceptions within marine mammalian ectoparasites, which are either not insects or maybe not part of an evolutionary arms race, we slightly rephrased the first sentence.	1 f.

When the results of pressure measurements are presented, the value does not have a unit, making it difficult to interpret the meaning.	Response: Referring to Leonardi et al. 2020 who performed these pressure measurements, we added a "*" to make it more comprehensible that kg*cm⁻² is the suitable pressure unit and additionally added the calculated pressure value in kPa.	55
The main statement in the introduction suggests that lice are host-dependent because they feed on blood. However, lice are permanent and obligate ectoparasites, regardless of whether they feed on blood or dead cells, feathers, etc. I think this is actually the fundamental point about lice dependence on the host: they cannot survive outside of it. For seal lice, this is more extreme because they infest amphibious hosts.	Response: We completely agree with the reviewer that the seal lice are obligate ectoparasites of seals, which cannot survive without contact to their hosts. We now emphasized this fact more detailed.	60 ff.
In Material and Methods it would be useful to provide the number of lice collected and analyzed under each method.	Response: We appreciate this suggestion and added the number of lice collected and analyzed under each method.	96, 110, 174
The discussion should include at least one paragraph analysing what this means for lice. I suggest discussing Bush, S. E., Sohn, E., & Clayton, D. H. (2006). Ecomorphology of parasite attachment: experiments with feather lice. Journal of Parasitology, 92(1), 25-31.	Response: Thank you for this suggestion. We implemented this publication in our discussion (lines 346 ff.) and discussed it shortly in the context of the mentioned hair diameters of seal hair and the claw morphology of the seal louse. However, we did not include a whole new paragraph for it, since it fitted in the context of the paragraph starting in lines 339 ff in our opinion much better.	347 ff.

Reviewers' comments:

Reviewer #1 (Remarks to the Author):

Dear authors,

Overall the text has a very clear structure and lacks excessive or confusing information. The introduction provides the reader with a good background on the sucking lice. The technical part of the project is well documented and the discussion provides an excellent explanation of the actual meaning of the experiment results.

Please find below some suggestions to improve the manuscript (with line numbers):

25: Please change "strongest insect attachment forces" to "strongest relative insect attachment forces" (larger insects could have higher absolute forces).

35: The statement about the successfulness of insects is subjective, since it lacks a clear criterion. Any group that is more inclusive (Arthropoda, Metazoa, etc.) has a higher species richness and a larger number of inhabited habitats. I would suggest to rephrase this sentence.

51: Please use dashes instead of hyphens for number ranges throughout the whole manuscript.

125: Please consider adding a more precise description how the false color image was composed. Which bandwidths translate to the RGB colour channels? (reproducibility)

193: Since the statistical analysis is entirely independent from the graphical user interface, R Studio does not need to be cited.

Apart from those minor corrections and suggestions, I highly recommend this manuscript for publication and I think that it is a very sound and valuable contribution to the study of insect biomechanics.

Reviewer #2 (Remarks to the Author):

The authors present, for the first time, an analysis of the features associated with the attachment of a seal louse to its host hair. Considering the great adaptive pressures on seals during the colonisation of the ocean by their pinniped hosts, this result is a key process. To understand the structure and how it works, they combine different techniques such as CLSM, SEM, and histology. The results show that the lice achieve an extraordinary attachment force. In this sense, the article could be a major contribution to understanding the adaptations of this particular group of insects. It is undoubtedly a valuable contribution, analysing a key process for this particular group of animals. I especially want to recognize the great work done with the figures. They are very clear and illustrative.

My main concern with this study relates to the fact that human hair was used for the experiment instead of seal hair. In the discussion, the author refers to the attachment to a seal capillary fibre. I think this is something that should be discussed. As the authors point out, seal lice have specific adaptations, many of which are very specific. So it would be possible that the diameter of the fibre is precisely related to the size or closure system of the claws. I suggest that this point be discussed.

Major comments:

- I found the title "ectoparasitic seal louse" redundant. By definition, lice are ectoparasites. The first sentence of the abstract postulates a co-evolutionary arms race between marine mammals and their parasites. However, this is not true for all types of parasites in these hosts. Moreover, there are no

insect parasites in cetaceans or sirenians. It would be easier to follow if the paragraph began with a reference to pinnipeds.

- When the results of pressure measurements are presented, the value does not have a unit, making it difficult to interpret the meaning.
- The main statement in the introduction suggests that lice are host-dependent because they feed on blood. However, lice are permanent and obligate ectoparasites, regardless of whether they feed on blood or dead cells, feathers, etc. I think this is actually the fundamental point about lice dependence on the host: they cannot survive outside of it. For seal lice, this is more extreme because they infest amphibious hosts.
- In Material and Methods it would be useful to provide the number of lice collected and analyzed under each method.
- The discussion should include at least one paragraph analysing what this means for lice. I suggest discussing Bush, S. E., Sohn, E., & Clayton, D. H. (2006). Ecomorphology of parasite attachment: experiments with feather lice. *Journal of Parasitology*, 92(1), 25-31.